# Priority Activities in Child and Adolescent Tuberculosis to Close the Policy-Practice Gap in Low- and Middle-Income Countries

**DOI:** 10.3390/pathogens11020196

**Published:** 2022-02-01

**Authors:** Karen du Preez, Betina Mendez Alcântara Gabardo, Sushil K. Kabra, Rina Triasih, Trisasi Lestari, Margaret Kal, Bazarragchaa Tsogt, Gantsetseg Dorj, Enkhtsetseg Purev, Thu Anh Nguyen, Lenny Naidoo, Lindiwe Mvusi, Hendrik Simon Schaaf, Anneke C. Hesseling, Andrea Maciel de Oliveira Rossoni, Anna Cristina Calçada Carvalho, Claudete Aparecida Araújo Cardoso, Clemax Couto Sant’Anna, Danielle Gomes Dell’ Orti, Fernanda Dockhorn Costa, Liliana Romero Vega, Maria de Fátima Pombo Sant’Anna, Nguyen Binh Hoa, Phan Huu Phuc, Attannon Arnauld Fiogbe, Dissou Affolabi, Gisèle Badoum, Abdoul Risgou Ouédraogo, Tandaogo Saouadogo, Adjima Combary, Albert Kuate Kuate, Bisso Ngono Annie Prudence, Aboubakar Sidiki Magassouba, Adama Marie Bangoura, Alphazazi Soumana, Georges Hermana, Hervé Gando, Nafissatou Fall, Barnabé Gning, Mohammed Fall Dogo, Olivia Mbitikon, Manon Deffense, Kevin Zimba, Chishala Chabala, Moorine Penninah Sekadde, Henry Luzze, Stavia Turyahabwe, John Paul Dongo, Constantino Lopes, Milena dos Santos, Joshua Reginald Francis, Magnolia Arango-Loboguerrero, Carlos M. Perez-Velez, Kobto Ghislain Koura, Stephen M. Graham

**Affiliations:** 1Desmond Tutu Tuberculosis Center, Department of Paediatrics and Child Health, Stellenbosch University, Cape Town 8000, South Africa; hss@sun.ac.za (H.S.S.); annekeh@sun.ac.za (A.C.H.); 2Pediatric Tuberculosis Working Group, Ministry of Health, Brasilia 70304-008, Brazil; betinamalcantara@gmail.com (B.M.A.G.); dearossoni@gmail.com (A.M.d.O.R.); anna.carvalho@ioc.fiocruz.br (A.C.C.C.); claudetecardoso@id.uff.br (C.A.A.C.); clemax01@gmail.com (C.C.S.); danielle.dellorti@saude.gov.br (D.G.D.O.); fernanda.dockhorn@saude.gov.br (F.D.C.); iliana.romero@saude.gov.br (L.R.V.); fatimapombo09@gmail.com (M.d.F.P.S.); 3Brazilian Network of Tuberculosis Research, REDE TB—Rede Brasileira de Pesquisas em Tuberculose, Rio de Janeiro 21941-909, Brazil; 4Department of Pediatrics, All India Institute of Medical Sciences, New Delhi 110029, India; skkabra@hotmail.com; 5Faculty of Medicine, Public Health and Nursing, Universitas Gadjah Mada, Yogyakarta 55281, Indonesia; rina_triasih@yahoo.com (R.T.); trisasilestari@gmail.com (T.L.); 6Menzies School of Health Research, Charles Darwin University, Darwin, NT 0810, Australia; josh.francis@menzies.edu.au; 7National Department of Health, Port Moresby 131, Papua New Guinea; margaretkal1978@gmail.com; 8Mongolian Coalition Against Tuberculosis, Ulaanbaatar 210648, Mongolia; bazra.ts@outlook.com; 9Tuberculosis Surveillance and Research Department, National Center for Communicable Diseases, Ulaanbaatar 210648, Mongolia; dorjgantsetseg1127@gmail.com; 10Tuberculosis Clinic, National Center for Communicable Diseases, Ulaanbaatar 210648, Mongolia; enhuush0425@gmail.com; 11Woolcock Institute of Medical Research, Ha Noi 100000, Vietnam; thuanh.nguyen@sydney.edu.au; 12Health Department, Cape Town 8000, South Africa; Gengiah.Naidoo@capetown.gov.za; 13National TB Control & Management Cluster, National Department of Health, Pretoria 0187, South Africa; lindiwe.mvusi@health.gov.za; 14Laboratory of Innovations in Therapies, Education and Bioproducts, Oswaldo Cruz Institute, Oswaldo Cruz Foundation, Rio de Janeiro 21045-900, Brazil; 15National Tuberculosis Program, Ha Noi 100000, Vietnam; nguyenbinhhoatb@yahoo.com; 16National Pediatric Hospital, Ha Noi 100000, Vietnam; phuc.h.phan@nhp.org.vn; 17International Union Against Tuberculosis and Lung Disease, 75001 Paris, France; arnauld.fiogbe.consultant@theunion.org (A.A.F.); gisele.badoum.consultant@theunion.org (G.B.); aouedraogo.consultant@theunion.org (A.R.O.); adjicomb@yahoo.fr (A.C.); albert.kuatekuate.consultant@theunion.org (A.K.K.); aboubacar.magassouba.consultant@theunion.org (A.S.M.); alphazazi.soumana.consultant@theunion.org (A.S.); georges.hermana.consultant@theunion.org (G.H.); nafissatou.fall.consultant@theunion.org (N.F.); fall.dogo.consultant@theunion.org (M.F.D.); olivia.mbtikon.consultant@theunion.org (O.M.); deffenseman@gmail.com (M.D.); kgkoura@theunion.org (K.G.K.); steve.graham@rch.org.au (S.M.G.); 18National Tuberculosis Program, Cotonou 03 BP 2819, Benin; affolabi_dissou@yahoo.fr; 19Faculty of Health Sciences, University of Abomey-Calavi, Cotonou 03 BP 2819, Benin; 20Health Sciences Unit, University Joseph Ki-Zerbo, Ouagadougou 03 BP 7047, Burkina Faso; 21Ministry of Health National Tuberculosis Program, Ouagadougou 03 BP 7047, Burkina Faso; tandaogo@yahoo.fr; 22National Tuberculosis Program, Yaoundé BP 6000, Cameroon; anniebisso@pnlt.cm; 23National Tuberculosis Program, Conakry 63570, Guinea; adabangou@yahoo.fr; 24National Tuberculosis Program, Niamey 22 646, Niger; 25National Tuberculosis Program, Bangui BP 729, Central African Republic; gahe_gi@yahoo.fr; 26National Tuberculosis Program, Dakar 12000, Senegal; docteurbg2000@yahoo.fr; 27National Tuberculosis Program, Lomé BP 526, Togo; 28Lusaka Provincial Health Office, Ministry of Health, Lusaka 10101, Zambia; drkevinmzimba@gmail.com; 29Department of Pediatrics and Child Health, School of Medicine, University of Zambia, Lusaka 10101, Zambia; cchabala@gmail.com; 30Children’s Hospital, University Teaching Hospitals, Lusaka 10101, Zambia; 31National Tuberculosis and Leprosy Program, Kampala 7025, Uganda; moorine.sekadde@gmail.com (M.P.S.); luzzehenry@gmail.com (H.L.); turyahabwestavia@gmail.com (S.T.); 32The Union Uganda Office, Kampala 7025, Uganda; jpdongo@theunion.org; 33National Tuberculosis Program, Ministerio da Saude, Dili NM 87109, Timor-Leste; costa_tb@yahoo.com; 34Hospital Nacional Guido Valadares, Ministerio da Saude, Dili NM 87109, Timor-Leste; milenalaysantos@yahoo.com; 35Colombian Association of Pediatric Pulmonology, National University of Colombia, Bogota 500001, Colombia; MagnoliArango@hotmail.com; 36Division of Infectious Diseases, University of Arizona College of Medicine, Tucson, AZ 85721, USA; cmperezvelez@gmail.com; 37COMUE Sorbonne Paris Cité, Faculté des Sciences Pharmaceutiques et Biologiques, Université Paris Descartes, 75006 Paris, France; 38École Nationale de Formation des Techniciens Supérieurs en Santé Publique et en Surveillance Epidémiologique, Université de Parakou, Parakou 03 BP 351, Benin; 39Department of Pediatrics, Murdoch Childrens Research Institute, University of Melbourne Royal Children’s Hospital, Melbourne, VIC 3052, Australia

**Keywords:** tuberculosis, child, adolescent, national tuberculosis program

## Abstract

Over the past 15 years, and despite many difficulties, significant progress has been made to advance child and adolescent tuberculosis (TB) care. Despite increasing availability of safe and effective treatment and prevention options, TB remains a global health priority as a major cause of child and adolescent morbidity and mortality—over one and a half million children and adolescents develop TB each year. A history of the global public health perspective on child and adolescent TB is followed by 12 narratives detailing challenges and progress in 19 TB endemic low and middle-income countries. Overarching challenges include: under-detection and under-reporting of child and adolescent TB; poor implementation and reporting of contact investigation and TB preventive treatment services; the need for health systems strengthening to deliver effective, decentralized services; and lack of integration between TB programs and child health services. The COVID-19 pandemic has had a significant negative impact on case detection and treatment outcomes. Child and adolescent TB working groups can address country-specific challenges to close the policy–practice gaps by developing and supporting decentral ized models of care, strengthening clinical and laboratory diagnosis, including of multidrug-resistant TB, providing recommended options for treatment of disease and infection, and forging strong collaborations across relevant health sectors.

## 1. Introduction

Over 60 years ago, frustration was first expressed with the persistent neglect of tuberculosis (TB) in children, defined as <15 years of age, particularly with the missed opportunity to prevent severe disease and mortality with isoniazid [1]. It was not until 50 years later that the Stop TB Strategy was launched in 2006 by the World Health Organization (WHO) and Stop TB Partnership as a new global strategy that focused on the needs of “both adults and children” irrespective of sputum smear status committing to “address TB/HIV, MDR-TB and *other* challenges”, including TB in “prisoners, refugees and *other high-risk* groups” [2]. In the same year, the WHO Stop TB Partnership Child TB Sub-group of the DOTS Expansion Working Group published the first guidance for National Tuberculosis Programs (NTP) on the management of TB in children [3]. The guidance included a recommendation that all children should be managed as part of routine NTP operations. It is only in the last decade that NTPs in TB endemic countries have begun to provide greater attention to child TB, with some milestones highlighted in the WHO’s recent Global TB Report, shown in Figure 1 [4]. 

A collective “Call to action for childhood TB” followed the first international child TB meeting, held in March 2011, which precipitated efforts to strengthen national surveillance data, as well as advocacy [5,6]. The first global estimates for TB in children (<15 years) were published in the 2012 WHO Global TB Report, a roadmap for child TB was published by multiple international stakeholders in 2013, and the second edition of the WHO guidance for NTPs was published in 2014 [7,8]. The critical importance of national data for child and adolescent TB was recognized, and, with time, the WHO recommended age-disaggregated reporting of all TB case notifications, including clinically diagnosed pulmonary TB and extrapulmonary TB [4,9]. The WHO’s End TB Strategy, that explicitly included children to a greater extent than previous global strategies, was launched in 2015, as were the first child-friendly and appropriately dosed fixed-dose combinations for the treatment of disease and infection in young children [10,11]. This led to regional consultations that brought together representatives of NTPs and child health within TB endemic countries, and to a greater recognition than previously of the need for cross-cutting collaboration between TB and child health services with the establishment of child TB working groups by NTPs [12]. At this time, the specific needs of adolescents (10–19 years) with TB began to be recognized, with the first estimates of age-disaggregated data [13]. The Child TB subgroup was upgraded to the Child and Adolescent TB Working Group, and the international roadmap was updated to include adolescents in 2018 [14]. 

TB in children and adolescents is a global health priority, with more than 1.5 million TB children and adolescents (0–19 years) developing TB each year [4]. Children represent 11% of the global TB burden but 16% of deaths, and the majority of the estimated 226,000 TB-related deaths in children each year are either not detected or not reported. The proportion of total TB caseload reported in 2019 that were children is shown by country in Figure 2 [15]. In 2021, the WHO reviewed recent evidence to inform new recommendations [16]. The updated and consolidated WHO guidelines on the management of TB in children and adolescents are now published, complemented by an operational handbook to aid implementation [17,18].

This perspective article aims to provide a series of viewpoints that specifically identifies priority actions to address the policy–practice gap in TB endemic countries and global regions. Indicators relevant to detection, treatment, and prevention of TB in children, as reported for the recent WHO Global TB Report by countries [4] represented in this article, are listed in Table 1. 

## 2. Overview of National Perspectives

A range of national and regional perspectives from TB endemic settings are presented by health workers and national leaders involved in providing TB care for children and adolescents. Table 2 provides a summary of current national programmatic initiatives or activities that are specific for child and adolescent TB. Efforts to address the policy–practice gap at the national level have increased and are variable between countries. Until recently, many NTPs had very limited policy guidelines relating to children, and they did not have national child TB Working Groups. Attention to the needs of adolescents is even more recent, if at all occurring previously [19]. Diagnostic approaches follow similar principles [8], but adaptation and implementation vary by setting and facility level. Countries, such as Papua New Guinea and Brazil, utilize home-grown scoring systems. However, the various approaches often include investigations, such as the tuberculin skin test, that are no longer widely available. 

Most countries now procure the dispersible, fixed-dose formulations of first-line drugs [11]. While progress is being made in the development of more child-friendly, appropriately dosed second-line drugs, the treatment of multidrug-resistant (MDR) TB remains challenging. The recent update by the WHO to support the use of the new drugs, bedaquiline and delamanid, for all ages is a major step forward toward all-oral regimens for children, thereby negating the need for injectables [16,17]. Increasing the coverage of household contact management with integrated, decentralized models of care is now a major focus, greatly enabled by having a range of safer and shorter regimens recommended for TB preventive treatment (TPT) [20]. However, the programmatic systems to record and report the main indicators of household screening coverage, along with the initiation and completion of TPT in eligible contacts, are still being established. Greater age-disaggregation of programmatic data through the child and adolescent age groups is hugely important given the major variations in epidemiological risks, diagnostic patterns, and treatment outcomes from birth until early adulthood. This is now requested for the WHO report and facilitated by the increasing use of electronic, rather than paper-based, surveillance systems. 

### 2.1. Brazil: Progress and Challenges in the Control of TB in Childhood and Adolescence

Brazil is currently listed by the WHO as a high-burden country for TB and TB/HIV, despite the implementation of TB control activities for over 50 years. Intradermal BCG vaccination of infants and young children up to five years old was introduced in the 1970s, as was “short-course” treatment for new cases. The dispensing of medications exclusively in public health services with obligatory notification allowed standardization of treatment, aggregation of data by the Ministry of Health and limited the risk of emergence of drug-resistant TB. In the 1980s, the Unified Health System, or Sistema Único de Saúde (SUS), was implemented to guarantee free healthcare, and, with the advent of the HIV epidemic, this included preventive measures and the distribution of antiretroviral (ARV) drugs. For the last 20 years, TB case notification has been completed through the Notifiable Diseases Information System, or Sistema de Informação de Agravos de Notificação (SINAN), which has strengthened surveillance and provided epidemiological data for monitoring and evaluation. National programmatic recommendations for TB are periodically updated and the NTP manual has included chapters that relate to children and adolescents since 2002.

Children (0–9 years) and adolescents (10–19 years) accounted for 1.7% and 6.4%, respectively, of the total 68,706 TB cases reported in Brazil in 2020 [21]. The proportions are lower than expected, especially in children, suggesting under-detection and/or under-reporting. Xpert MTB/RIF was introduced in 2013 and Xpert MTB/RIF Ultra in 2019. There is likely to be under-detection with bacteriological and clinical diagnosis in adolescents, as well, as suggested by a recent retrospective study of a vulnerable population with poor living conditions in Rio de Janeiro [22]. Since 2011, WHO-recommended treatment regimens for adolescents and children have been applied, and child-friendly dispersible combined formulations are now available. Treatment outcomes for the 2019 cohorts of child and adolescent cases were: treatment success rates of 71% and 75%; deaths 3.8% and 1.5%; and lost to follow-up were 8.7% and 12.5%, respectively [21]. Risk factors for unfavorable outcomes in adolescents have been identified, such as extreme vulnerability and HIV infection [23]. Brazil is not a high-burden country for MDR TB, but this is also under-detected. From 2016 to 2020, only 0.2% of MDR-TB cases occurred in children (0–9 years), and 5.4% in adolescents (10–19 years) [21]. The new second-line oral drugs of bedaquiline and delamanid are available.

The WHO has recently set targets for implementation of TPT, primarily among young child contacts and people living with HIV [4,15]. In 2018, an information system for TB infection (TBI) notification and TPT was implemented with adoption by all federated units in 2021. The COVID-19 pandemic compromised contact screening coverage, which fell from 63% in 2019 to 55% in 2020 [21]. However, TBI notifications increased during this period, with children and adolescents accounting for 25.5% of cases receiving TPT. Children and adolescents in Brazil participated in landmark international trials that provided evidence of effectiveness and safety of shorter TPT options now recommended by the WHO [24,25]. These TPT options of 4R (daily rifampicin for 4 months) and 3HP (once weekly rifapentine and isoniazid for 3 months) are now available in Brazil, as is the interferon-gamma release assay to test for infection.

The Ministry of Health created a “child TB working group” in 2017. This is comprised of specialists in child and adolescent TB, and a major task was to develop strategies that would enable the decentralization of diagnosis and treatment to the level of the Primary Health Care (PHC) Units. This strategy utilizes a child TB clinical diagnostic scoring system developed by the Ministry of Health, in 2002, and has since been validated [26]. The health worker is provided with guidance on further investigation and decisions to treat for TB at a decentralized level of care [27].

Despite the aforementioned efforts and leadership within the Ministry of Health, TB is still neglected in children and adolescents in Brazil. One major challenge is that the health system for children and adolescents is not well adapted to provide TB care. There is a low implementation of policy guidelines, as well as available tools for detection and prevention. It is necessary to give more attention and voice to children and adolescents, and to listen to them, providing a model of care where they feel welcomed and that takes into account epidemiological, anthropological, social, and economic aspects [28,29]. Only then will it be possible to replace the suffering imposed by TB with a future that seeks to eliminate this disease.

### 2.2. India: Child and Adolescent TB Challenges and Way Forward

The India TB report estimated that 342,000 children in India developed TB in 2019, accounting for 31% of the global burden of child TB, and 13% of the national TB burden [30]. To improve the care and prevention of TB in children and adolescents, major changes have recently been made by the National TB Elimination Program (NTEP) of government that include: improved diagnostic algorithm facilitating use of rapid molecular tests and detection of drug-resistant TB in children, introduction of child-friendly drug regimens, pediatric-oriented monitoring, and updated options for TB preventive treatment [31].

Diagnosis is a major challenge. To improve case detection and treatment, the NTEP has introduced a new diagnostic algorithm so that chest X-ray (CXR) is an initial, routine investigation in children with or without history of TB contact who have TB-related symptoms, such as persistent unremitting cough, weight loss, or failure to gain weight. Clinical diagnosis is still important due to the paucibacillary nature of disease in children. To improve yield of bacteriologically confirmed TB in children, the NTEP now recommends use of molecular tests (such as Xpert MTB/RIF or Truenat TB test) in preference to smear examination in pulmonary, as well as extrapulmonary TB [27]. The sensitivity of molecular diagnostics varies from 15% to 30% in tertiary research settings, to less than 10% under programmatic conditions [32,33]. Of a total of 94,415 presumptive TB cases in Indian children presenting to public and private clinics in ten cities in India in which free-of-charge upfront Xpert MTB/RIF was used, Xpert was positive in 6.6% compared to 2% on smear microscopy for acid-fast bacilli (AFB) [33]. Around half of the specimens tested were non-sputum, and nearly 90% of positive cases commenced treatment on the day of diagnosis. Such an approach is associated with a considerable increase in health system costs.

All children diagnosed with TB are registered on an electronic portal called “NIKSHYA” that helps with tracking all children to reduce treatment interruptions [31]. There have been major changes to the recommended treatment regimens for children. There is now a single regimen for all types of newly diagnosed TB: rifampicin, isoniazid, pyrazinamide, and ethambutol, during intensive phase, with three drugs (rifampicin, isoniazid, and ethambutol) recommended in the continuation phase due to a high background prevalence of isoniazid resistance. To improve treatment completion rates, a pill box for each child containing a full course of treatment is kept at TB treatment centers from treatment initiation. To identify drug-resistant TB in children, an integrated diagnostic algorithm has also been developed [34]. In the study of upfront Xpert MTB/RIF use, 545 (8.7%) of 6270 cases with *Mycobacterium tuberculosis (M.tb.)* complex detected by Xpert were rifampicin-resistant [33]. All diagnostic services are provided free through the NTEP laboratory network [30].

To improve acceptability of national guidelines, it has been important to engage all stakeholders who manage child TB in development of new guidelines and ensure that resources are available to them for treatment of children with TB. This process involved experts and academics and included members of the Indian Academy of Pediatrics. This is a nationally representative organization that signed a memorandum of understanding with the Central TB Division in October 2019 to build public and private sector capacity, which included district level training to strengthen management and notification of child TB cases [30].

TPT is an important NTEP priority, and guidelines for implementation were recently published [30,35]. TPT is now offered to children of all ages who are household contacts of bacteriologically confirmed pulmonary TB patients. Widespread implementation of this activity is being done with support from the general health system. However, there is much to be done to increase coverage. India reported that only 42% of young children (<5 years) who were household contacts of TB source cases received TPT in 2019 [15]. For child TB index cases, reverse contact tracing to identify the source of infection is also recommended [35]. 

In summary, there is now strong political will to support implementation of updated NTEP policy guidelines and reach national targets. The major challenges include decentralization of services for detection and prevention, improving case finding utilizing the resources available through the NTEP, and scaling up contact screening and TPT. 

### 2.3. South Africa: Key Programmatic Opportunities and Challenges to Improving TB Services for Children and Adolescents 

South Africa is among the countries with the highest burden of TB globally, with an estimated total TB incidence of 328,000 (95% CI 230,000–444,000) in 2020 (Table 1) [15]. The incidence among children under 15 years of age is estimated at 30,000 (95% CI 19,000–41,000). As much as there has been a gradual decline in TB notifications since 2009 [36], the notification gap remains high (16,442 in children <15 years). TB has been integrated into the Integrated Management of Childhood Illness package of care for young children since 1998. To prioritize and increase the focus on children, a separate Childhood TB guideline was developed in 2013. At a policy level, child and adolescent TB services are well integrated with other health programs and services. However, despite substantial investment, capacity building, and guideline and policy development, implementation remains sub-optimal and varies substantially across provinces, resulting in missed opportunities for prevention, detection, and treatment. In 2020, South Africa reached 66% of the child TB case finding target agreed upon at the United Nations High Level Meeting for TB, and only 35% of the target for TPT initiation in eligible child contacts below 5 years [37].

Whilst TPT has been implemented for many years in young and HIV-positive children who are contacts of people diagnosed with infectious TB, reporting on TPT implementation only started in 2017. Many missed opportunities for the prevention of childhood TB remain, and more effort is required in screening child contacts and increasing TPT uptake. A recent review estimated that less than a third of eligible child contacts initiated TPT, and less than a third of those who initiated, completed six months of isoniazid [38]. Qualitative research identified challenges at various levels but specifically noted a lack of training for healthcare workers and inadequate systems to monitor TPT delivery [38,39]. TPT completion is not monitored and reported, but plans are currently in place to include these in the National Indicator Data Set in 2022. Given the poor TPT completion rates, the anticipated revision of national TPT guidelines to include shorter regimens, such as 3 months of daily isoniazid and rifampicin (3HR), recommended by the WHO [20], is eagerly anticipated, while access to 3 months of once weekly rifapentine and isoniazid is not yet available. BCG vaccination at birth is national policy, and, though BCG supply has been restored following recent global shortages, challenges remain in uptake and coverage.

Analysis of routine South African surveillance data for drug-susceptible TB from 2004 to 2016 found that TB case notification rates had declined the least in young children (<5 years) living with HIV and in older adolescents (15–19 year of age), irrespective of HIV status [40]. Therefore, TB control efforts should increase attention to these vulnerable sub-groups. Many young children with uncomplicated TB are still diagnosed at hospital-level, while a diagnosis could feasibly be made at the PHC level [41,42]. Regular training for PHC workers providing child and maternal health and TB services will increase their confidence and competency to diagnose TB in children [43]. TB meningitis continues to be a common and preventable cause of death or major long-term disability in children [44]. Initial symptoms are non-specific, resulting in missed opportunities for earlier diagnosis, especially at the PHC level [45], while substantial opportunities for prevention through BCG vaccination and TPT remain. In rural or remote areas, access to microbiological testing and X-ray services remains a challenge, resulting in delayed presentation and diagnosis of TB. Even in urban areas, where these services are more readily accessible, interpretation of CXRs for child TB [46] and limited attempts to obtain samples for microbiological testing at PHC level remain major challenges to TB diagnosis, especially in young children. Bacteriological confirmation is especially important in children at risk of drug-resistant TB in this setting. 

Of 729,463 children and adolescents routinely registered and treated for drug-susceptible TB in South Africa during 2004–2016, treatment success (cured or treatment completed) was reported in only 81%, and 116,808 (16%) were lost to follow-up during treatment [47]. Mortality whilst on treatment declined during this period to 2% in 2016. However, the highest mortality was observed in children below 2 years of age and among adolescent females [47]. With close observation, individualized care, and global leadership in expertise and experience, children with drug-resistant TB in South Africa have excellent outcomes, with treatment success in over 90% [48].

Despite the availability, since 2016, of child-friendly fixed-dose combination formulations for first-line treatment in young children, South Africa only introduced these in mid-2021 due to substantial delays in regulatory approvals. While South Africa has made great advances in adopting decentralized models of care for children with drug-resistant TB, child-friendly formulations are not yet available for many second-line drugs. Therefore, treatment remains complicated by a substantial pill burden, especially for those living with HIV. The shorter and injectable-free drug-resistant TB treatment regimens now allow for decentralized home-based care and a family-centered approach to treatment and monitoring. Bedaquiline and delamanid have been available for children since 2018, although with some age restrictions in younger children. 

South Africa has used electronic case-based TB surveillance data since 2004 and, over the past five years, has invested in strengthening health data systems, including those for TB surveillance. However, more comprehensive indicators and integration of multiple data sources are needed for adequate monitoring and evaluation of child and adolescent TB care. Surveillance should include age-disaggregated data on prevention, TB disease spectrum, and post-TB morbidity and should be available to inform improvement of services at facility level. Strengthened information systems between hospitals and PHC facilities to improve linkage to care for children and adolescents can further improve outcomes [49,50,51,52,53]. With the TB surveillance system as it is now, it is impossible to effectively monitor program performance due to lack of access to the system at district, province, and national levels, where data management and the use of data for action is compromised. A system for program monitoring, interoperable with the NHLS Laboratory information system and death notification system, is urgently required in South Africa.

The impact of COVID-19-related restrictions on TB services in South Africa has been substantial, with a 26% decrease in Xpert tests performed in 2020 [54]. No data have been published yet on the impact of COVID-19 on child and adolescent TB services, but a dramatic decline (almost 50%) in bacteriologically confirmed TB has been observed in children presenting to hospitals in the Western Cape province (unpublished data presented at The Union conference, in 2021, by HS Schaaf et al.). 

The actual burden of TB in children and the case notification gap are unknown since the national TB prevalence survey excluded children. Quantifying the pediatric burden more accurately using tools appropriate for children needs to be prioritized in the investment case to ensure that the limited resources are efficiently and effectively directed for impact. Establishing a national “Child and Adolescent TB Working Group”, e.g., as part of the multi-sectoral South African TB Think Tank [55], which advises the NTP to advocate for the needs of children and adolescents in South Africa will be an important step toward addressing some of these translational challenges. One of the first tasks of such a working group should be to update the national pediatric TB guidelines (2013) to align with updated WHO recommendations (2021) and operational guidance for implementation. Comprehensive and regular training for all healthcare workers working in TB and in maternal, child, and adolescent health services could greatly improve integration and care. Decentralized models of care for treatment of TB disease and TPT, such as through home-based care or ward-based outreach, could provide more family-centered options to effectively prevent, diagnose, and treat TB in children and adolescents. Active engagement with civil society organizations will help further create awareness and should involve child and adolescent TB ambassadors and their families, who can be the voice of their peers and advocate for their specific needs. 

### 2.4. Indonesia: Challenges to TB Detection for Children in the Time of COVID

The COVID-19 pandemic is having a massive impact on health services in Indonesia. The first case of COVID-19 in Indonesia was detected in March 2020, and, as of end November 2021, a total of 4,256,409 people have had SARS-CoV-2 infection confirmed, and 143,830 have died [56,57]. The pandemic has impacted the utilization of healthcare services in Indonesia, with a 70% decrease in non-COVID-19 healthcare service delivery from January to September 2020 [58]. This abrupt reduction was mainly because most health facility resources were diverted to the COVID-19 response, people’s fear of exposure to SARS-CoV-2 at health facilities, and to the policy decision by clinic facilities to reduce service time and delay elective care [59].

Indonesia has the second highest number of TB cases annually in the world, as shown in Table 1. However, children and adolescents are often neglected in the response to TB control, with particular challenges for case detection highlighted by the difficulty to confirm the diagnosis microbiologically, as well as a major case notification gap, even when cases are diagnosed [60]. The COVID-19 pandemic has had a direct impact on TB control, including reversing recent efforts by the Indonesian NTP, to improve detection and treatment of child TB; the 70,341 reported child TB cases in 2019 was a 15% increase compared to 2018. However, only 32,930 child TB cases were reported in 2020, a 53% reduction from the previous year and a return to numbers reported in 2016, when around 35,000 child TB cases were reported. Of 34 provinces in Indonesia, only two (Papua and West Java) provinces successfully reached the target for child TB case finding. 

In the Mimika district of the high-incidence province of Papua, TB case finding activities was also affected by the pandemic. In 2020, the number of presumptive child TB cases fell by 38%, and child TB case notifications fell by 21%. However, the proportion of children among all presumptive TB and reported TB cases remained constant, at 12% and 21%, respectively. The number of children who died during TB treatment increased by 116% during the pandemic, which contributed to a fall in treatment success rate among child TB, from 77.5% in 2019 to 72.0% in 2020. It is plausible that co-infection with COVID-19 and delayed access to healthcare were contributing factors (unpublished data, T Lestari). 

Diagnosis remains a key challenge. While smear microscopy and/or Xpert MTB/RIF assay were performed in 54% of presumptive child TB cases, only 7% children treated for TB in 2020 were bacteriologically confirmed, and just 3.3% among young (<5 years) children (unpublished data, T Lestari). The challenges also affect detection of drug-resistant TB. Only 84 (1.1%) of 7961 bacteriologically confirmed drug-resistant TB cases in 2020 were children, and only 33 (39%) of them were enrolled on treatment [61]. Extrapulmonary TB constituted about 28% of all child TB cases in Mimika, with mainly clinical diagnosis due to limited health resources to collect and test appropriate specimens (unpublished data, T Lestari). Sputum induction was commonly used for diagnosis in a hospital setting, but this aerosol-generating procedure is associated with particular risk in the context of the COVID-19 pandemic. The use of stool Xpert has been recommended based on supportive evidence from a multicenter study conducted in Indonesia. However, uptake of this new recommendation is very limited, with training and supervision required. In those not bacteriologically confirmed, major challenges remain with clinical diagnosis. 

The last national child TB guideline was published in 2016 and is currently being updated. TPT is a key recommended strategy to prevent to TB disease in high-risk groups, such as young child (<5 years) contacts or people living with HIV of any age. Since 2020, the NTP has extended TPT recommendations to all eligible household contacts, including adolescents. Six months of isoniazid preventive therapy remains the most common TPT regimen in Indonesia, but, in 2021, the NTP began to provide shorter TPT regimens of 3RH and 3HP. The NTP has set a target to provide TPT to 49,222 young child contacts in 2020, which is 40% of the estimated eligible number. However, the coverage remains very low (Table 1). A project in the Mimika district, in the Papua province, started in 2017 [62], increased the uptake of TPT for young child contacts five-fold by 2019, but this has been negatively impacted by the COVID-19 pandemic. 

### 2.5. Papua New Guinea: Addressing Challenges in a High-Burden Child TB Country 

In Papua New Guinea (PNG), TB in children constitutes around a quarter of all TB cases reported to the NTP [63]. The reasons for such a high proportion compared to other countries is due to a number contributing factors that include population demographics (children make up a high proportion of the population); under-detection and reporting of adult TB; a high prevalence of risk factors for child TB, such as malnutrition and high rates of infection; and reliance on clinical diagnosis. Major challenges exist for bacteriological confirmation, and a lack of diagnostic skills in some health workers results in both overdiagnosis and underdiagnosis. Xpert is increasingly available in PNG but is still limited to the major provincial centers [64]. Currently, more than 95% of child TB is clinically diagnosed, and, even if respiratory specimens are available, the yield of Xpert for PTB is low. Many healthcare workers lack training in child TB management in PNG. Moderate to severe malnutrition is common in children with TB and adds challenges to diagnosis, as well as risk for worse outcomes. 

Treatment success for child TB remains below target, with more than 30% of child TB lost to follow-up [63]. Except for the National Capital District, provinces in PNG do not have specific clinics to follow up child TB, and caregivers are not properly supported. PNG was one of the first countries to introduce child-friendly fixed-dose formulations to improve treatment in young children managed at Port Moresby General Hospital [65]. Recent progress has also been seen in both diagnosis and treatment of child TB in Port Moresby, under a TB/HIV project which supports additional child TB focused human resources, tools, training and transport. Healthcare workers were trained and sensitized to detect child TB, perform gastric aspirate and other specimen collection for bacteriological diagnosis, treat patients, and provide treatment support. This improved model of child TB care needs to be expanded to other provinces. Due to the geography of PNG being mountainous with many islands, decentralization of health services to the provincial level has long been recognized as necessary. 

The COVID pandemic has greatly affected delivery of TB services due to restricted travel and reduced access to health facilities, including shutdowns of clinics or repurposing of services, such as human resources and deploying Xpert for SARS-CoV-2 diagnosis. As a result, child TB notifications have fallen, while treatment interruptions and loss to follow-up have increased.

Priorities to improve child TB response are identified in the National TB Strategy and include increasing case detection and susceptibility testing for all children with TB, increased treatment success for all child TB, and improved programmatic response for child TB. Addressing these priorities will be achieved through interventions that expand and decentralize activities for diagnosis, treatment, contact evaluation, and prevention. These will include procurement of diagnostic equipment; training for healthcare workers, including on the use of gastric aspirate, fine-needle aspiration, and other procedures; and updating and dissemination of all guidelines, algorithms, and standard operating procedures. Interventions to increase treatment success and prevention for child TB include: expansion of child TB treatment, including child-friendly formulations and all-oral regimens for drug-resistant TB; contact evaluation and TPT through community TB workers with implementation of integrated patient and family education and counseling secessions [66]; expansion of treatment facilities with nutritional support for all child TB patients; and identification and training of a provincial child TB focal person to support training of healthcare workers on child TB management. Additional Global Fund support provides transport and food incentives for children and families of drug-resistant TB patients.

Interventions to strengthen technical and managerial capacity for child TB includes the recruitment of a national child TB focal person who will liaise with PNG Pediatric Society and will identify and train child TB focal persons based at the provincial level. The PNG Pediatric Society and pediatricians in the National TB Technical Working Group have long supported NTP activities. PNG has approved and updated child friendly-treatment regimens for TB treatment and TPT and has recently contributed to the Guidelines Development Group involved in the recent update of the WHO child and adolescent TB guidelines. 

### 2.6. Mongolia: Major Challenges for TB Services, Including for Children and Adolescents

TB services for children and adolescents in Mongolia face a wide spectrum of challenges at all levels. These include a lack of capacity at the ground level to volatile political landscapes at the leadership level. Mongolia is a large, sparsely populated country, and health services for TB are overly centralized to the capital, Ulaanbaatar, while the rest of the country is relatively neglected. There is a widespread shortage of trained and capable staff to provide TB care and services. According to 2015 NTP reports, about 50% of personnel working in TB were already at retirement age [67]. Existing staff are overloaded with excessive workload, despite being in a high-risk and low-paying job. There are inadequate policies and a lack of coordinated actions at the levels of planning, training, working conditions, and performance management. Mongolia has recently been included in the WHO list of 30 high-burden MDR TB countries, in 2021 [15], with notification rates remaining very low—at about 30% of those estimated in the last decade [67]. A national survey reported that the majority (68%) of TB patients in Mongolia experience “catastrophic” treatment costs, and the individual, family, and health systems costs are even greater for patients with MDR TB [68]. The challenges are currently magnified by the COVID-19 pandemic, with active case finding activities decreasing fourfold, almost 2% increase in TB mortality in 2020 compared to 2019 [69].

The Mongolian NTP has restructured with six different managers in the last decade, and there are only two specialized pediatric TB clinicians nationwide. Decision-making is greatly limited by frequent changes in leadership, which hinders political will and commitment to achieve programmatic goals. Leadership issues have recently been highlighted by the inability to keep pace with rapidly changing WHO TB guidelines. In response to the shortage of competent healthcare workers for TB services, the government of Mongolia shortened the duration of TB residency training from two years to nine months, resulting in an influx of TB clinicians. In 2018–2020, the TB project of the Global Fund has supported 28 clinicians to specialize in TB, and, of these, 89.3% are already working in TB. However, the abbreviated training does not include any or sufficient time allocated to training in child TB. Human resource policies and training need to address long-term needs, and pediatricians need to be included in trainings on TB. 

TB infection is common in Mongolian children; exposure in households and schools is common [70,71]. A pilot project on screening and management of household contacts of MDR TB cases in 19 sites of Mongolia demonstrated that 17 (5%) of 349 contacts developed TB (unpublished data, B Tsogt), higher than reported to the NTP in 2019 [67]. It has also shown that there is a disconnect between TB and child health services, as well as a major lack of confidence and competence in the diagnosis of TB in children. District TB clinicians who are expected to provide services for child TB should receive training that would enable every TB dispensary to have at least one trained TB clinician to overcome the diagnostic challenges in children. The current shortage of child TB expertise leads to delays in diagnosis and treatment initiation, as well as a lack of implementation of household screening and TB prevention. NTP data in 2020 [69] reported that 76% of child TB was lymph node TB, which highlights the difficulties that health workers are having to diagnose the more common pulmonary TB. On the other hand, there is also overdiagnosis with CXR in healthy children with a positive TST [69]. In the context of screening and management of household contacts of people with MDR TB, this has been noted to lead to unnecessary hospitalization and treatment of children with second-line TB medications.

### 2.7. Vietnam: Where Are the Child TB Cases?

Vietnam is listed as one of the world’s 30 high-burden TB countries, with almost 100,000 people reported with TB in 2020, of an estimated 172,000 incident cases [4]. According to the Vietnam NTP, there should be between 10,000 and 20,000 child TB cases annually. The annual risk of TB infection in school-aged children was previously estimated to be 1.7%, although more recent evidence from a mainly rural setting suggests that this is lower in young children [72,73]. However, only 1.5% of reported TB cases were children (<15 years); of the 1404 cases in 2020, two-thirds were older children (5–14 years). The proportion of child cases of MDR/RR TB was even lower, at 17 cases or 0.5% of all MDR TB in 2020. Therefore, the diagnosis and notification of pediatric TB remain significant challenges in the country. 

Neonatal BCG vaccination is included in the Expanded Program on Immunization, and coverage has been at a high level since 2009. In 2019, 95% of neonates in Vietnam received BCG, though vaccination rates were lower for children who were born in rural areas, in ethnic minority communities, or to mothers with low education [74]. The high BCG coverage will impact on severe disease in young children, but clinical diagnosis is an ongoing challenge as a reliance on bacteriological confirmation results in under-detection, especially among high-risk young children [75]. Of pulmonary TB cases in children, the sputum smear-positive cases were mostly in older children. Though Xpert is now recommended for bacteriological confirmation in children, access is limited. 

The NTP’s child TB Working Group has been expanded to include new international partners. The group has recently developed and completed guidelines on childhood TB management, training, and communication materials. There is a lack of knowledge on child TB diagnosis among healthcare staff, as well as inadequate collaboration between child health and TB services. Most TB hospitals do not have a pediatric department. In 2020, the NTP organized training courses for health workers on the diagnosis and management of childhood TB for 63 provinces. The network that connects the National Pediatric Hospital with provincial hospitals has been strengthened for direct or online joint diagnosis consultation and referral. In 2020, a pilot study on Xpert testing of stool in children was started. Treatment regimens are consistent with WHO guidelines. The rate of treatment completion or cured among all treated children was 67%. It was higher for those evaluated as treatment cohort in 2019, but around 30% of children were not evaluated for a treatment outcome. 

A national guideline on TPT implementation was revised in 2020, providing guidance for contact investigation and TPT among young child (<5 years) contacts or children living with HIV without TST administration. In 2020, the NTP organized training courses for health workers on TBI management for children in 25 high-burden provinces. After ruling out active TB, eligible children are offered either 6H or 3RH as TPT, while children aged 2 years or older have an additional option of 3HP. The NTP has supported the roll-out of a community-based contact management program since 2012. In the second half of 2020, 3552 young child contacts were initiated on TPT, which represents 54% of 6617 eligible children in the pilot program. However, the overall coverage of TPT for young child contacts in 2020 reported to the WHO was only 5% [15]. There is an ongoing challenge of limited access to TST tests among children 5 years and older for TBI diagnosis, under-notification of TPT coverage, uptake, and completion.

### 2.8. French-Speaking African Countries: A Focus on Child Contact Management 

TB contact screening and management is a key strategic intervention in the fight against TB. This intervention aims to include active case-finding among contacts with prompt treatment to improve outcomes and reduce ongoing transmission with the provision of TPT to eligible contacts to reduce risk of TB disease. While this strategy has been recommended for many years, using at least six months of isoniazid preventive therapy, it is rarely implemented in French-speaking African countries. There is evidence from a randomized trial on the safety of four months of daily rifampicin that included children from Guinea and Benin [25]. A recent project supported by The Union demonstrated high uptake and completion, as well as effectiveness and safety, of a shorter TPT regimen using the child-friendly fixed-dose formulation of rifampicin-isoniazid in urban settings in four countries [76].

The Contributing to the Elimination of Tuberculosis in Africa (CETA) project is now being implemented within routine activities by NTPs in eight francophone African countries (Benin, Burkina Faso, Cameroon, Guinea, Niger, Central African Republic, Senegal, and Togo) with support by The Union. The first component of this project enabled the implementation of TB contact investigation with a focus on several vulnerable groups, including young child contacts. At the beginning of the CETA project, 150 Basic Management Units were identified as pilot sites in the eight countries. The tools to implement the activity were developed, and a national training was organized for nurses and community healthcare workers. Preliminary results report that, between October 2020 and March 2021, 4540 home visits were carried out, 2940 children under 5 years were initiated on TPT, and 89 were treated for active TB (CETA Activities Report 2021, unpublished data, KGK).

Project implementation has identified several major challenges. Despite the training, some countries note a low level of involvement and proficiency of community healthcare workers. Some countries experience a high turnover of trained staff. There are difficulties in effectively referring children to evaluate for TB disease before initiation of TPT. There are particular difficulties of access to populations who live in areas considered unsafe. The project has also highlighted the lack of coordination between child health and TB services, as well as the refusal of some parents to agree to TPT for their healthy child. Finally, the COVID-19 pandemic has reduced attendance to clinics for referral assessment or follow-up [77].

In order to scale up this project activity, several actions are necessary that can be quickly implemented. The national policy on community healthcare workers should be reviewed and updated to include TB contact investigation in the package of the activities of community healthcare workers. The capacity of healthcare providers to detect and manage child TB needs strengthening, with collaboration between the NTP and pediatric services. The development of a communication and advocacy plan is required to improve community education and engagement, and financing, including that for home visit costs, could be included in Global Fund grants. 

In terms of future perspectives, the implementation of a simple clinical algorithm suitable for use by community healthcare workers and nurses would help to detect or eliminate active TB in child contacts. The activity should be digitalized in real time, and the use of digital chest X ray with computer-aided detection should be evaluated [77]. Many countries recommend a test for infection before TPT initiation, as does the WHO, for older child contacts who are HIV-negative; therefore, a rapid test for infection would be a real advantage. Finally, there are currently no interventions that specifically support adolescents with TB in French-speaking African countries. This component remains to be developed and should be prioritized by NTPs in future grants.

### 2.9. Zambia: Challenges for NTP and Action to Strengthen Child and Adolescent TB Services

Zambia has a high prevalence of tuberculosis and is listed as a high-burden country for TB, TB/HIV, and MDR/RR (Table 1) [4,78]. Of the estimated nearly 60,000 annual TB cases, a third are undiagnosed or unreported. The case detection gap is higher among children, contributing only 6% of the annual caseload, while coverage for contact screening and TPT for young child contacts remains a challenge [79]. TB/HIV coinfection rates are high in adults and children, but there has been a sustained improvement in TPT coverage for people living with HIV. The burden of MDR TB in children remains poorly undefined. 

Key national priorities for addressing child TB include: addressing the case detection gaps, sustained improvement in treatment outcomes, rolling out TB prevention services, and addressing the threat posed by the COVID-19 pandemic. Tackling the case detection gap requires addressing the perennial diagnostic challenges in child TB, which restricts case detection to district, regional, or tertiary facilities, thereby limiting access to care. Low notification rates have been attributed to low index of suspicion, underdiagnosis, and under-reporting of TB in children within the health system [80]. Autopsy studies of Zambian children conducted over a decade apart have shown that TB is persistently a frequent cause of death in children with respiratory illness that is not often detected antemortem [81,82]. Therefore, implementing systematic screening and creating clinical awareness with emphasis on the use of clinical tools to improve case detection and early treatment is a priority to reduce TB-related morbidity and mortality in Zambian children. 

The NTP will require investing more in healthcare worker capacity building, as well as improving access to diagnostic tools. This includes access to old tools, such as CXR, as well as the adoption of new tools, that can be used at lower-level facilities, such as the use of Xpert in stool or detection of LAM in urine for target populations, to tackle the case detection gaps of both drug-susceptible and drug resistant TB. Improving access to treatment through decentralization of care, ensuring availability of child-friendly formulations, and strengthening linkage to HIV care for children living with HIV are all essential elements to sustain good treatment outcomes and achieving the End TB targets. 

The implementation and scale-up of prevention, through increasing coverage and completion of TPT for eligible child TB contacts, is another key priority. Low community awareness, healthcare provider perceptions, supply chain problems, and policy implementation gaps have been recognized as barriers that must be overcome for successful IPT implementation in the country [79,83]. To address these policy–practice gaps, the NTP has provided policy guidance on the systematic implementation of TPT, while setting coverage targets for child contacts and PLWH, ensuring provision of commodities, adopting shorter TPT regimens, addressing health worker and community barriers, providing trainings and mentorship of healthcare providers, and engaging civil societies. 

The COVID-19 pandemic threatens to derail gains made in the provision of TB services. Measures introduced to contain COVID-19 had an impact on the delivery of all public health services, including TB care [84]. This has necessitated the adoption of approaches that minimize the challenges that people with TB face in accessing services and care [85]. Measures implemented to mitigate the impact of COVID-19 include providing national guidance on the maintenance of TB services, adapting procedures to avoid, where possible, frequent contact with health services, including telephone follow-up, co-messaging of TB and COVID-19, and the use of community agents to support care. The NTP adopted the use of a virtual platform to hold weekly meetings that linked all health facilities in the country. This allowed tracking of performance of TB services against targets and helped in identifying areas for support that were being impacted during COVID upsurges.

Sustained political will is critical to address the identified priorities. The NTP has updated its National TB Strategic Plan to align with the Global End TB goals to capture and address child TB interventions. This has been an inclusive process that engaged all key stakeholders, including the National TB/HIV technical working group, the childhood TB technical working group, civil societies, non-governmental organizations, and other partners, involved in TB services. The outputs of the National Strategic Plan aim to address some of the policy gaps in adolescent TB, integration with maternal and child health services, TB/HIV services, improving access to treatment, and sustaining good treatment outcomes.

### 2.10. Uganda: Addressing Programmatic Challenges to Improving TB Services for Children and Adolescents

The WHO lists Uganda as a high-burden country for TB and TB/HIV. Over recent years, the country has documented a steady improvement in child and adolescent TB case finding (currently at 12% of the total case load) and treatment coverage (Table 1) with support from key stakeholders. Despite significant investment into in-service capacity building, gaps remain that emphasize the need to evaluate and review implementation to address the “know-do gap”. There are opportunities that can be leveraged to bolster the delivery of quality patient-centered TB services for children and adolescents [86]. 

There have been recent efforts to develop training, tools, and job aides to support the lower-level health facilities and community health workers, which resulted in increased case detection, improved treatment outcomes, and improved access to TPT for contacts [86,87]. However, updated information on the programmatic management of TB in children has not been widely disseminated, including within referral hospitals, training institutions, and the private sector. In addition, there is limited access to chest radiography, often due to non-functionality or out-of-pocket cost for the service. 

Due to diagnostic delays, children with TB, especially those living with HIV, commonly present with complicated TB, partly contributing to sub-optimal treatment outcomes. Dispersible child-friendly formulations are now available for fixed-dose combinations of first-line drugs, except for dispersible ethambutol (100 mg). Case notification of child TB is inconsistent, and treatment follow-up is particularly challenging, for children within the migrant populations, such as pastoralists. There are major gaps in detecting and managing drug-resistant TB due to limited health worker capacity and few options for child-friendly formulations of second-line drugs.

Implementation of TB contact evaluation has been inconsistent due to limited dissemination of the approved operational guidelines, lack of tools, and insufficient focused funding support. Health workers often lack the clinical skills or confidence to exclude TB among contacts before providing TPT. Concerns about potential toxicity and resistance development have hindered TPT uptake, especially among under-five contacts. This is coupled with the general community perception that TB does not affect children, contributing to a low demand for the service. 

There is an ongoing need to build capacity, including equipping health facilities, training, mentorship, and supervision. The DETECT child TB project showed that effective training and health worker support to overcome these misperceptions and build confidence and capacity increased case detection more than two-fold [87]. The model is currently being scaled up to 50 districts. The NTLP transitioned to using age disaggregated data for child TB reporting and recently initiated the phased introduction of e-case based surveillance for TB. Once scaled up, this provides an opportunity for real-time reporting; the current coverage is less than 10% of the TB treatment facilities. 

Challenges within the private sector are the limited engagement of clinicians and that the costs of diagnostics are not affordable for the majority of the population. Many private facilities have transitioned to electronic medical records, yet paper-based documentation is the main platform for the national health information system. The limited access to a sample transportation system, coupled with delays in sample pick-up and delivery of results, are barriers to the use of Xpert MTB/RIF for the private facilities.

The new patient-centered National Strategic Plan, and TB Multisectoral Accountability Framework, provides opportunities for enhancing child TB service delivery, including adaptation and scale-up of successful innovations. While the policy favors the clinical diagnosis of TB, especially among young children, TB services must be further decentralized with support for screening, treatment availability, and follow-up at the peripheral facility level (level II), and with linkages up to the referral, specialist facility (level V). Further, there is currently inconsistent targeted mentorship and supervision, partly due to limited resources, such as limited numbers of skilled mentors. The delay in finalizing and disseminating the updated school health policy has hindered the implementation of standardized TB-focused interventions in schools. In Uganda, the school health policy is housed under the education sector, which calls for enhanced approaches to engage the other key players. 

The COVID-19 pandemic has negatively impacted case detection. Many health workers have focused on symptom screening for COVID-19, and there was a heightened stigma for TB during the initial phase of the COVID pandemic. The “stay at home” message targeting people with respiratory symptoms, coupled with travel restrictions, meant that children with TB-related symptoms were more likely to be kept at home. To mitigate the negative impact of COVID on TB service delivery, the MOH/NTLP developed guidance to ensure continuity of care, which guided the implementation of innovations in collaboration with stakeholders. 

### 2.11. Timor-Leste: Recent Attention to Childhood TB

Timor-Leste has one of the highest TB incidence rates in the Asia-Pacific region, with an estimated annual incidence that has ranged between 498 and 508 per 100,000 population per year [15]. The prevalence of drug-resistant TB is low, with rifampicin resistance detected in only 1.3% of cases included in a recent drug resistance survey [88]. Under-detection of TB is a major challenge for all age groups, exacerbated by geographical challenges with remoteness and terrain, limited knowledge and understanding within the community, difficulties with access to radiologic and laboratory diagnostic testing, and variability in diagnostic approach across the country. More than 80% of TB patients in Timor-Leste experience catastrophic costs in relation to their TB diagnosis and care [89]. In 2020, only 51% of estimated incident cases were detected and started on treatment [15].

It is likely that there is a very high burden of TB in children and adolescents in Timor-Leste. Approximately 40% of the population is aged less than 15 years, and malnutrition is common, with approximately 49% of young children (<5 years) being moderately or severely stunted [90]. Anecdotal experience of pediatricians working in the national referral hospital (Hospital Nacional Guido Valadares) in Dili includes frequent encounters with severe manifestations of disseminated TB in children. However, only 8% of new and relapse TB case notifications in 2020 were in children aged less than 15 years, as shown in Table 1 [15]. Fewer cases are diagnosed in the 0–4-year age group than the 5–14-year age group. The proportion of reported TB cases that are children varies between municipalities, making up less than 5% of diagnosed cases in some settings.

Children living outside the capital Dili or other major centers are less likely to get access to chest x-ray or molecular testing with Xpert. While most cases are clinically diagnosed, there is a general reluctance to diagnose TB and commence treatment without laboratory confirmation and/or involvement of a pediatric specialist. Access to pediatricians outside of the six referral hospitals is limited, and there are systemic, as well as patient and family level, barriers to transferring patients to hospital unless they are acutely and severely unwell.

The NTP has led on initiatives to improve case detection of TB in adults and in children, supported by the WHO and other partners. Timor-Leste’s first Guidelines for the Management of Tuberculosis in Children were developed by the NTP in 2016 and were accompanied by a series of training events that included clinicians from all 13 municipalities of Timor-Leste. Xpert testing is recommended as the first-line laboratory test for children with presumptive TB. The recently revised Comprehensive TB Guidelines for NTP and the NTP’s National Strategic Plan for Ending TB 2020–2024 have prioritized increasing access to chest x-ray and Xpert for diagnosis, as well as the importance of supporting and empowering clinicians to make clinical diagnoses of TB in children. A national TB prevalence survey is planned for 2022, and, while it will not enroll children, it will provide the most accurate estimates to date of the burden of TB in Timor-Leste, which can be used to measure progress in case detection and treatment over the coming years.

The COVID-19 pandemic has significantly and negatively impacted care-seeking behavior in Timor-Leste, and TB case detection rates have dropped as a result. However, it has also led to a significant investment in increasing access to GeneXpert testing for SARS-CoV-2, TB, and other diseases, with additional GeneXpert machines installed, and plans for at least one machine to be installed in all 13 municipalities of Timor-Leste by 2022. The COVID-19 pandemic has also illustrated the importance and potential effectiveness of household contact evaluation for airborne infectious diseases, such as COVID-19 and TB. 

There is an urgent need to improve contact evaluation for TB cases in Timor-Leste, which can have an impact on increasing secondary case-finding and providing access to TPT. This is especially important for young child contacts, who frequently represent missed opportunities for the prevention of TB disease in Timor-Leste [91]. 

### 2.12. Colombia: Challenges and Priorities for Change to Improve the Care of Children and Adolescents with Tuberculosis

The number of cases and the quality of care for children and adolescents with TB in Colombia vary greatly by region due to socioeconomic disparities. Healthcare services in rural areas are precarious. Allotted budgets are insufficient and often further reduced by the corruption prevailing in the country [92]. The disarticulation of previously vertical public health programs (including TB) is a consequence of administrative decentralization and of the municipalization of health programs [93]. Additional barriers include bureaucracy related to health insurance programs, as well as fragmentation of care [94]. Physical access to healthcare facilities is often limited due to the presence of illegal armed groups. As a result of decades of violence in rural areas, Colombia has the most internally displaced persons in the world, estimated at 8.3 million individuals, which is more than 16% of the national population. In addition, 1,731,000 Venezuelan refugees have migrated to Colombia [95]. Furthermore, expertise in child and adolescent tuberculosis is limited, and the retention of trained healthcare professionals is poor due to job insecurity and poor salaries. Finally, TB is still highly stigmatized in Colombia, with considerable discrimination toward sufferers.

The aforementioned challenges demand the strengthening of both the NTP and local programs. These should be better integrated with other child health programs and should work in coordination with various entities to guarantee children’s rights to healthcare. Furthermore, TB care teams need to be trained (and to receive continuing education) in child TB, to have job security (to ensure continuity), to have adequate resources to implement current standards of care, and to work in partnership with local communities. 

In Colombia, the incidence rate of TB has been increasing, estimated to be 37 cases per 100,000 population for 2020 [15]. Only 35% of estimated child TB cases are detected, and even less during the COVID-19 pandemic; 328 cases were reported in children in 2020 compared to 404 cases in 2019. To assure timely TB care through and after the COVID-19 pandemic, it will be necessary to effectively support local TB control programs to achieve more opportune active case finding amongst both adults and children, as well as more timely notification of detected cases.

Delayed diagnoses of TB in children and adolescents are not uncommon [96]. There is limited detection of TB disease at outpatient clinics offering primary care. Reasons include limited knowledge of TB and misdiagnosis with other diseases, which leads to a high percentage (58%) of cases being diagnosed at the time of a hospitalization. When TB is clinically suspected in a child, the guidelines of the NTP recommend CXR, a test for infection, and to seek bacteriological confirmation [97]. However, the availability of laboratory diagnostics is inadequate, especially at PHCs, and, even when available, results are delayed. Consequently, presumptive diagnoses are often made based on clinical, radiological, and epidemiological criteria. In children with paucibacillary disease, treatment is often not initiated until the disease is advanced and/or complications have developed. Treatment success was reported to have been attained in 78.3% of children, but sequelae are not reported. There are difficulties with treatment adherence and support, with follow-up especially challenging in communities that are geographically isolated. Some indigenous peoples reject treatment for cultural reasons. 

To overcome these challenges, there should be no barriers to access comprehensive care for early diagnosis of TB exposure, infection, or disease, especially in high-risk vulnerable populations. It is necessary to shorten the time intervals between the onset of symptoms, diagnosis, and initiation of treatment. This requires NTP to provide technical assistance to local TB care teams and to decentralize laboratory capacity with molecular diagnostics. The NTP must assure uninterrupted availability of TB medications without barriers, the mutual commitment between the NTP and the patient that takes into account cultural aspects and allows for adequate treatment support. Lastly, the support of civil society organizations is required to improve treatment administration and adherence, especially in the indigenous population.

Contact screening is implemented in only a minority of infectious TB patients, and TPT for eligible at-risk children is rarely initiated or completed. There is limited understanding of TBI amongst healthcare professionals and laypeople, compounded by a low perception of the risk in children. Some communities and parents are reluctant to have their otherwise healthy children evaluated for TB exposure. Testing for TBI is very limited with persistent shortages of tuberculin and a scarcity of personnel trained in tuberculin skin testing. Interferon-gamma release assays are only available in large cities and are very expensive. According to the NTP, in 2020, TB contact investigations resulted in 1,448 children receiving TPT for confirmed or presumptive TBI; only 185 (12.8%) were less than 5 years of age. Access to testing for TBI and CXR is required [98]. A family- and community-based approach needs to be developed in order to promote awareness of TBI and the value of prevention with TPT regimens that are shorter than isoniazid preventive treatment.

## 3. Common Challenges and Way Forward

Attention to the needs of children with TB disease or with TB infection who live in TB-endemic, often resource-constrained settings has increased over the last decade, and, in recent years, adolescents have also received a greater and deserved focus [6,19]. Yet, despite the increased advocacy, greater inclusion in guidelines and policy and research, and the availability of child-friendly prevention and treatment, it is estimated that more than 650 children still die of TB every day [4]. Therefore, TB is an important cause of morbidity and mortality in children and adolescents globally, despite being treatable and preventable. 

The country perspectives in this article provided by national pediatric TB experts from TB endemic countries all identify the notification gap as a challenge—missed opportunities to detect and treat, as well as under-reporting of child and adolescent TB, who are diagnosed. Effective first-line treatment is widely available and effective, including child-friendly dispersible formulations, but the challenge is to make the diagnosis. Identified underlying drivers contributing to under-detection were the shortage of trained staff, lack of access to diagnostic tools, reluctance to diagnose TB without bacteriological confirmation, and limited access to expertise at central hospitals. Child TB services and expertise tend to be strongly centralized, but most children and adolescents requiring TB treatment or TPT do not present, at least initially, to the specialized services, nor would most require referral if services were more decentralized [99]. All child TB working groups now recognize that strengthening diagnosis (clinical and bacteriological) is critical and needs to happen at all facility levels. The diagnosis and treatment of MDR TB in young children is consistently identified as a major challenge at all levels of the health system. 

Despite almost universal recommendations for decades, implementation of TPT for child contacts remains well below targets [15]. Countries have variably adopted recently recommended TPT options (Table 2), but a huge policy–practice gap for the implementation of contact investigation is recognized. There is often a lack of integration between TB and child health services [14], which is now recognized by many as important to address in order to improve services for detection, treatment, and prevention. All countries have reported a significant negative impact of the COVID-19 pandemic on TB case detection and treatment outcomes, including increased mortality and loss to follow-up, as well as on services for prevention. 

The importance of accurate surveillance data for both child and adolescent TB is recognized as critical [9]. Countries are increasingly able to provide age-disaggregated data and are introducing programmatic indicators for TPT along the cascade of care (Table 2). The need for data for real-time program monitoring and evaluation has been highlighted by multiple countries. Despite the availability of national guidelines and child-friendly treatment options in many countries, implementation of services remains sub-optimal due to inadequate training and surveillance systems, with high turnover of trained staff and low demand noted as underlying factors.

The WHO has published updated and consolidated guidelines for NTPs on the management of child and adolescent TB, and the guidelines are supported by an operational handbook [17,18]. A number of the recommendations are directly relevant to the aforementioned needs: decentralized models of care, treatment decision approaches that can be used at all levels of care, strengthening laboratory diagnosis, including of MDR/RR TB, and updated treatment options, including those for TB meningitis and MDR TB, as well as including the recent TPT guidelines with dosage charts. The WHO guidelines and handbook are complimented by the timely report of the Defeat Childhood TB project, a multi-country assessment of childhood TB programm funded by the Elizabeth Glazer Pediatric AIDS Foundation and Unitaid, that calls national policymakers and world leaders to action [100]. The report outlines a framework consisting of seven key policy recommendations for NTPs to strengthen child and adolescent TB services (Figure 3). 

## Figures and Tables

**Figure 1 pathogens-11-00196-f001:**
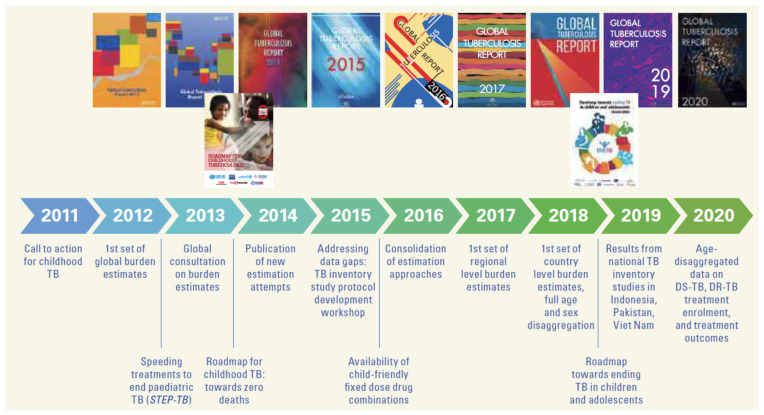
Global milestones related to TB in children and adolescents, 2011–2020. Reprinted from Ref. [4].

**Figure 2 pathogens-11-00196-f002:**
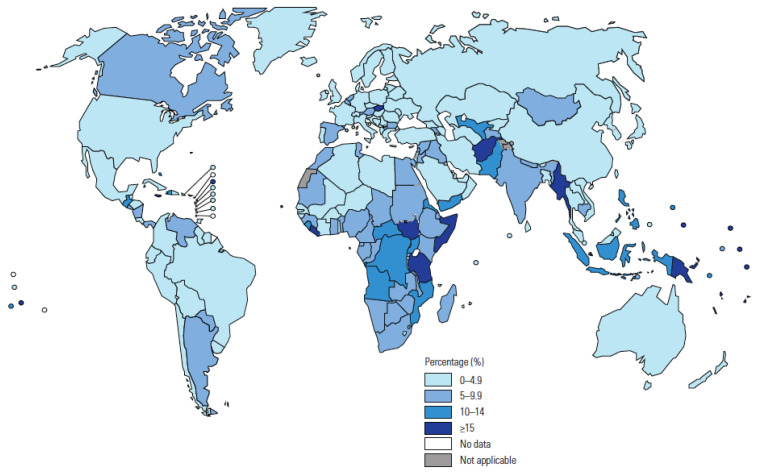
Percentage of new and relapse TB cases that were children (aged <15 years) in 2019. Reprinted from Ref. [4].

**Figure 3 pathogens-11-00196-f003:**
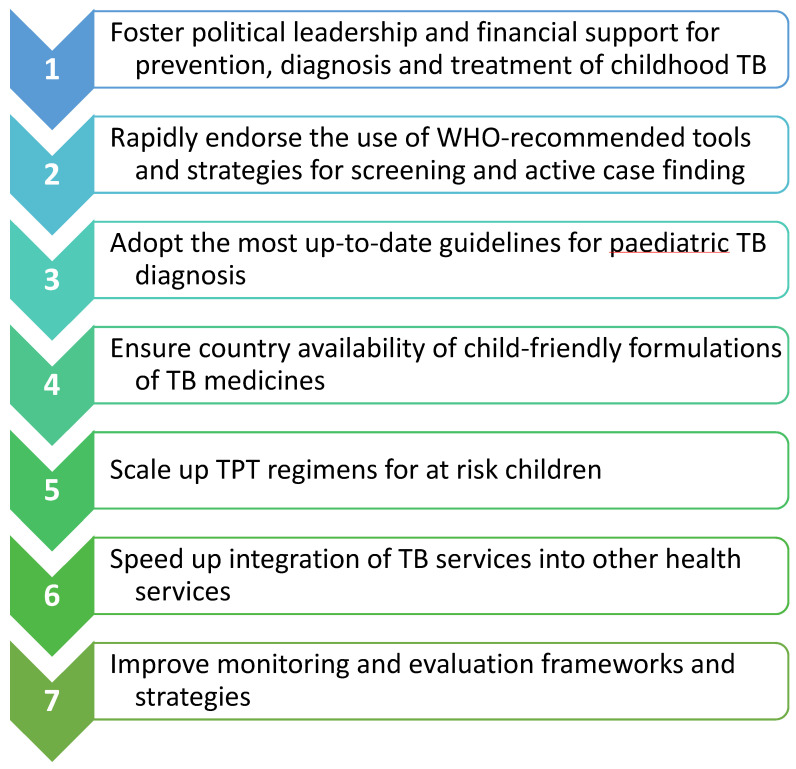
Seven key policy recommendations for NTPs as listed in the “Call to action to DEFEAT childhood TB.” Adapted from Ref. [100]. These actions are consistent with the roadmap [14] and can guide child TB working groups to define the next steps to address remaining country-specific challenges to address the policy–practice gaps that this article highlights by providing detailed perspectives and country-specific updates from a wide range of settings, for the first time, on efforts to address child and adolescent TB.

**Table 1 pathogens-11-00196-t001:** Data reported by countries in this review as included in the current 2021 WHO Global TB Report [15].

Country	WHO High-Burden Country List Inclusion	Total TB Incidence 2020 Data	Treatment Coverage, 2020 % Notified of Estimated	Proportion of Notified TB Cases Are Children (<15 years)	Treatment Success Rate 2019 Cohort	TPT Coverage for Eligible Young Child (<5 Years) TB Contacts
Benin	-	6700	58%	5%	89%	NR
Brazil	TB; TB/HIV	96,000	78%	3%	69%	59%
Burkina Faso	-	9600	59%	3%	81%	24%
Cameroon	TB/HIV	46,000	48%	5%	86%	43%
Central African Republic	TB; TB/HIV	26,000	48%	13%	81%	16%
Colombia	-	19,000	64%	3%	75%	38%
Guinea	TB/HIV	23,000	66%	6%	89%	NR
India	TB; TB/HIV; MDR/RR-TB	2,590,000	63%	6%	84%	42%
Indonesia	TB; TB/HIV; MDR/RR-TB	824,000	47%	9%	83%	4%
Mongolia	TB; MDR/RR-TB	14,000	27%	12%	88%	8%
Niger	-	20,000	56%	4%	83%	NR
Papua New Guinea	TB; TB/HIV; MDR/RR-TB	39,000	72%	22%	73%	23%
Senegal		20,000	65%	5%	91%	33%
South Africa	TB; TB/HIV; MDR/RR-TB	328,000	58%	7%	79%	51%
Timor Leste	-	6700	48%	8%	91%	NR
Togo	-	3000	79%	3%	87%	NR
Uganda	TB; TB/HIV	90,000	68%	12%	82%	34%
Vietnam	TB; MDR/RR-TB	172,000	58%	1%	91%	5%
Zambia	TB; TB/HIV, MDR/RR-TB	59,000	68%	6%	89%	28%

**Table 2 pathogens-11-00196-t002:** A summary of support for “child-friendly” tuberculosis management by region and country.

Country	Child TB Working Group	Specific NTP Guidelines *	Specific Inclusion of Adolescents	Diagnostic Approach or Algorithm	Child-Friendly Treatment Options	TPT Options **	Programmatic Indicators for Contact Management ^	Training Manual and Job Aides	Age-Disaggregated Data #
America Region									
Brazil	Yes	Yes	Partial	Yes	Yes	6H, 3HP, 4R	Yes	Yes	Yes
Colombia	No	No	No	Yes	Yes	6H, 3HP (HIV+)	Yes	No	Partial
Africa Region									
Benin	Yes	Yes	No	Yes	Partial	6H	Yes	No	No
Burkina Faso	No	Yes	No	Yes	Yes	3RH (HIV-) 6H (HIV+)	Yes	Yes	No
Cameroon	Partial	Yes	Yes	Yes	Yes	6H	Yes	Yes	Yes
Central African Republic	No	Yes	No	Yes	Yes	6H, 3RH	No	No	Yes
Guinea	Partial	Yes	Yes	Yes	Yes	6H	Yes	Yes	Yes
Niger	No	Yes	Yes	Yes	Yes	6H	Yes	Yes	Yes
Senegal	Yes	Yes	No	Yes	Yes	6H	Yes	Yes	Yes
South Africa	No	Yes	No	Yes	Partial	6H (3RH, 3HP) ^¥^	Yes	No	Yes
Togo	No	No	No	No	Yes	6H, 3RH	No	No	No
Uganda	Yes	Yes	Partial	Yes	Yes	3RH, 6H, 3HP	Yes	Yes	Yes
Zambia	Yes	Yes	Partial	Yes	Yes	6H, 3RH, 3HP	Yes	Yes	Yes
South-East Asia Region									
India	Yes	Yes	Yes	Yes	Yes	6H, 3HP	Yes	Yes	Yes
Indonesia	No	Yes	No	Yes	Yes	6H, 3RH, 3HP	Yes	No	Yes
Timor Leste	No	Yes	No	Yes	Yes	6H, 3RH	Yes	No	Yes
Western Pacific Region									
Mongolia	Partial	Yes	No	Yes	Yes	6-9H, 3HP, 1HP	Yes	No	No
Papua New Guinea	No	Yes	No	Yes	Partial	6H, 3RH, 3HP	Partial	No	Yes
Vietnam	Yes	Yes	No	Yes	Yes	6H, 3RH, 4R, 3HP	Yes	Yes	Yes

* Guidelines as stand-alone manual or chapter within National TB Program (NTP) guidelines; ** These TB preventive treatment (TPT) options are for contacts of drug-susceptible TB cases, N.B. some NTPs, such as Mongolia and Indonesia, also have TPT guidelines that include levofloxacin for contacts of multidrug-resistant TB cases; # Able to report notifications and treatment outcomes by age groups of 0–4 years, 5–9 years, 10–14 years, and 15–19 years; ^ Programmatic indicators of coverage, TPT uptake and TPT completion. ^¥^ 3RH and 3HP are included as planned TPT options for children in South Africa. 6H = 6 months isoniazid; 3HP = 3 months isoniazid and rifapentine; 4R = 4 months rifampicin; 3RH = 3 months rifampicin and isoniazid. Green: fully implemented; Orange: partially implemented; Red (pink): not implemented.

## Data Availability

Not applicable.

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
