# Peer review of "Priority Activities in Child and Adolescent Tuberculosis to Close the Policy-Practice Gap in Low- and Middle-Income Countries"

_pathogens, 2022, doi:10.3390/pathogens11020196_

Round 1

Reviewer 1 Report

As an ancient infectious disease, TB has threatened human’s health and lives for more than 7000 years. Until now, TB prevention and control are still facing challenges in some low- and middle-income counties, especially in extremely underdeveloped countries. This review gave a highlight on the policy-practice gap in low- and middle-income countries and overarching challenges in some terms among 19 TB endemic low and middle-income countries, which provides a series of viewpoints that specifically identifies priority actions to address the policy-practice gap in TB endemic countries and global regions. Overall, I am glad to review this paper which gives a reminder to the public that some strategies must be done to reduce the mortality and morbidity of TB in children. However, some issues should be revised before publication.

  1. The biggest problem of this review is the inaccuracy of data citation. For example, the author claimed that “Children represent 11% of the global TB burden but 15% of deaths” in line 123, however, the Global TB Report 2021 released by WHO showed that, among HIV-negative people, children represent 11% of the global TB burden but 16% of deaths (aged <15 years) in 2020.
  2. There are mistakes between the main text and the cited references. For example, the author claimed that “Figure 2. Percentage of new and relapse TB cases that were children (aged <15 years) in 2019. This figure is reproduced from the Global TB Report 20204.”. In this case, reference 4 is “Global TB Report 2021. World Health Organization, Geneva, 2021”, but the figure is obtained from Global TB Report 2020. These errors may reduce the accuracy and rigour of the article and may send wrong information to the public. The authors must carefully check each data over through the whole text.
  3. It has been well known that China is the high TB burden country in the world, which accounts for 8.5% of TB burden in the world in 2020 (It is third behind India and Indonesia). But I have not found any data from China, authors should explain why China was excluded in this review.
  4. In the section of “2. Overview of National Perspectives”, the authors discussed the TB condition of children and adolescent TB in each country. However, the sub-headings are not unified and appear messy. It is suggested that the author include the name of the country discussed in each sub-heading so that the logical structure will be better.
  5. The resolution of Figure 1 is very poor, authors should enhance it by redrawing a new one.
  6. Some data of this manuscript should be updated to 2021, such as the percentage of new and relapse TB cases that were children in 2019 in Figure 1. As far as I know, WHO has released the latest Global Tuberculosis Report 2021 in Oct 2021.
  7. Table 1, Total TB incidence 2020 data should be shown as incidence rate per 100,000 population.
  8. Table 2, some acronyms should be explained in the notes below the table, such as 6H, 3HP, and 4R.
  9. The style of this paper should be changed from “Article” to “Review”.

Author Response

attached response to  reviewer.

Reviewer 2 Report

Some minor comments are as below.

  1. You may merge 2.3, 2.4, and 2.5 into one under South Africa to harmonize with other countries (one chapter per country).
  2. If there are some common components (such as epidemiology, national strategy, diagnosis, treatment, prevention, etc.) in order, followed by specific situations per country description, it would be easier to compare countries.

Author Response

response to  reviewer attached

Round 2

Reviewer 1 Report

The authors have modified the manuscript following my comments, and the manuscript looks very good in this revision. I would like to suggest to publish this paper in the journal.